# DNA-Binding One Finger Transcription Factor *PhDof28* Regulates Petal Size in Petunia

**DOI:** 10.3390/ijms241511999

**Published:** 2023-07-26

**Authors:** Yuanzheng Yue, Wuwei Zhu, Huimin Shen, Hongtao Wang, Juhua Du, Lianggui Wang, Huirong Hu

**Affiliations:** 1College of Landscape Architecture, Nanjing Forestry University, Nanjing 210037, China; zhuwuwei@njfu.edu.cn (W.Z.); shm12345shm@163.com (H.S.); w917631006@163.com (H.W.); jh.du@siat.ac.cn (J.D.); wlg@njfu.com.cn (L.W.); 2Co-Innovation Center for Sustainable Forestry in Southern China, Nanjing Forestry University, Nanjing 210037, China; 3Shenzhen Institute of Advanced Technology, Chinese Academy of Sciences, Shenzhen 518000, China; 4Key Laboratory of Horticultural Plant Biology, Ministry of Education, College of Horticulture and Forestry Sciences, Huazhong Agricultural University, Wuhan 430070, China

**Keywords:** ornamental plant, Dof transcription factor, petal elongation, feedback regulation

## Abstract

Petal size is a key indicator of the ornamental value of plants, such as *Petunia hybrida* L., which is a popular ornamental species worldwide. Our previous study identified a flower-specific expression pattern of a DNA-binding one finger (Dof)-type transcription factor (TF) *PhDof28*, in the semi-flowering and full-flowering stages of petunia. In this study, subcellular localization and activation assays showed that *PhDof28* was localized in the cell nucleus and could undergo in vitro self-activation. The expression levels of *PhDof28* tended to be significantly up-regulated at the top parts of petals during petunia flower opening. Transgenic petunia ‘W115’ and tobacco plants overexpressing *PhDof28* showed similar larger petal phenotypes. The cell sizes at the middle and top parts of transgenic petunia petals were significantly increased, along with higher levels of endogenous indole-3-acetic acid (IAA) hormone. Interestingly, the expression levels of two TFs, *PhNAC100* and *PhBPEp*, which were reported as negative regulators for flower development, were dramatically increased, while the accumulation of jasmonic acid (JA), which induces *PhBPEp* expression, was also significantly enhanced in the transgenic petals. These results indicated that *PhDof28* overexpression could increase petal size by enhancing the synthesis of endogenous IAA in petunias. Moreover, a JA-related feedback regulation mechanism was potentially activated to prevent overgrowth of petals in transgenic plants. This study will not only enhance our knowledge of the Dof TF family, but also provide crucial genetic resources for future improvements of plant ornamental traits.

## 1. Introduction

*Petunia hybrida* L. is an herbaceous flower in the family Solanaceae, with wide distribution in gardens due to its high ornamental and economic values. Petals are important flower organs that not only send strong and recognizable visual signals to attract pollinators [1], but are also vital ornamental organs in petunias whose sizes significantly influence the plant’s ornamental value. Thus, exploring the molecular regulatory mechanism of petunia petal size is crucial for the improvement of its ornamental and economic values.

After the emergence of petal primordium, subsequent petal development can generally be divided into petal cell division and cell expansion stages, which both involve numerous complex regulatory networks [2,3]. It has been demonstrated that endogenous plant hormones, such as gibberellin acid (GA), ethylene, indole-3-acetic acid (IAA), and jasmonic acid (JA), play critical roles in the determination of petal size by activating related signal transduction pathways, and the antagonistic effects between ethylene and GA has been well illustrated in the rose petal expansion process [4,5,6]. However, the molecular mechanism of other participating hormones in the petal expansion process is not well understood.

Transcription factors (TFs) are extensively involved in the regulation of plant growth and development. Recently, a number of TFs associated with petal cell division have been identified, and their underlying molecular regulation of petal proliferation have been verified [5,7]. However, while some petal expansion regulatory genes have been characterized, their roles in petal expansion remain to be elucidated. In *Arabidopsis thaliana* L., the BIGPETALp basic helix-loop-helix (*AtBPEp*) and auxin response factor 8 (*AtARF8*) TFs—which are both preferentially expressed in the petals—have been shown to interact and influence the petal size by inhibiting petal cell expansion, while the expression of *AtBPEp* was shown to be positively correlated with endogenous JA levels [7,8]. The nucleus-localized NAC TF *RhNAC100* and GRAS subfamily TF *RhGAI1* in *Rosa hybrida* L. was shown to repress the expansion of petal cells by directly inhibiting the expression of the cell expansion-related cellulose synthase *RhCesA2* gene [3,9]. A member of the teosinte proliferating cell factor (TCP) TF family, *CmTCP20*, exhibiting nuclear localization but with no transcriptional activation, was shown to promote *Chrysanthemum morifolium* Ramat. petal elongation/expansion by interacting with a key JA signaling pathway regulator *CmJAZ1-like* gene [10].

Members of the Dof TF family are plant-specific regulators, which contain a conserved Dof domain with C_2_-C_2_ single zinc finger structures at their N terminal. The conserved domain can both interact with proteins or bind to the *cis*-motif sequences on the promoters of target genes. Dof TFs have been reported to widely participate in the regulation of plant organ development [11]. In apples (*Malus pumila* Mill.), the expression level of *MdDof24* was shown to significantly increase during flower development [12]. In Moso Bamboo [*Phyllostachys edulis* (Carrière) J. Houz.], the expression profiles of *PheDof12*, *PheDof14*, and *PheDof16* were highly correlated with floral organ development, and the genes were predicted to play critical roles in the development of flowers [13]. Notably, the nucleus-localized *AtDof5.4* gene in *Arabidopsis* exhibited a higher expression level in the petals, and its overexpression could inhibit the proliferation and expansion of flower organ cells; similar functions were also detected for *AtDof4.1* [14,15]. However, the regulatory functions of Dof TF family members in petunia organ development have been rarely reported.

Our previous study on the petunia Dof TF family identified *PhDof28*, which showed specific expression in flower organs and association with petunia petal development [16]. The purpose of this study was to further functionally characterize *PhDof28* roles in petal cell expansion and supply gene resource for the petal size improvement breeding of ornamental plants. These results will potentially deepen the understanding of the Dof TF family’s functions in the regulation of petal expansion in plants.

## 2. Results

### 2.1. Sequence Features, Subcellular Localization, Self-Activation, and Expression of PhDof28

The PhDof28 sequence protein constituted 301 amino acids with a highly conserved C_2_-C_2_ domain (Figure 1A). A phylogenetic tree revealed that *PhDof28* could cluster well with other Dof family members from Solanaceae (Figure 1B). Subcellular localization analysis indicates that *PhDof28* was localized in the nucleus, while a transcriptional self-activation assay exhibited that pGBKT7::*PhDof28* could grow well on sd/-trp and sd/-trp-ade plates and turn the sd/-trp-ade culture medium containing x-alpha-gal blue, which indicated self-activation (Figure 1D). The results of qRT-PCR indicated that *PhDof28* was preferentially expressed in the top region of petals, with an up-regulated trend from the pre- to full-flowering stages (Figure 2).

### 2.2. Phenotypic Characteristics of Transgenic Petunia Plants

PCR detection was used to obtain seven transgenic petunia plants (Appendix A), of which three plants had bigger petals, and OE-6 line was selected for systematic phenotypic observation. Compared to the wild-type plant, transgenic petunia overexpressing *PhDof28* exhibited much larger flower diameter and a longer corolla tube (Figure 3A,B). The phenotype analysis of OE-6 petal cells at the top, middle, and bottom sections at the full-flowering stage using SEM (Scanning Electron Microcopy) showed that the top part of transgenic petunia petals had larger cells than that of wild type petals, and the cellular morphology of transgenic petals was relatively irregular (Figure 3C,D). However, no morphological differences were observed in the cells of the bottom part between wild-type and OE-6 petals (Figure 3E).

To test whether the enlarged petal phenotype could be stably inherited in the offspring, the F_1_ plants of OE-6 line, including OE6-3, OE6-5, OE6-6, and OE6-7 were selected for further phenotypic observations. The results showed that the phenotypes of F_1_ and F_0_ plants were similar, especially at the top part of the petals (Figure 4A), while the expression levels of *PhDof28* in the F_1_ petals at the pre- and full-flowering stages were significantly up-regulated compared to those of the wild-type petunia (Figure 4B), which indicated stable inheritance of the large petal phenotype in the transgenic petunia offspring.

### 2.3. Phenotypic Characteristics of Transgenic Tobacco Plants

A total of six positive transgenic tobacco plants were isolated (Appendix A), and OE-31 and OE-28 with bigger petals were selected for further phenotypic observation. Results showed that tobacco plants overexpressing *PhDof28* shared similar phenotypes with those of transgenic petunia (Figure 4 and Figure 5A,C), which confirmed that *PhDof28* overexpression could promote petal elongation and expansion. The SEM observation revealed that transgenic OE-28 and OE-31 tobacco plants had larger cells, which were more loosely arranged as compared to the wild-type at the top part of the petal (Figure 5B). Meanwhile, cells in the middle petal parts of the transgenic tobacco were wider and larger than those of wild-type petunia (Figure 5B), while cells at the bottom part of the petal of transgenic tobacco plants were much longer than those of the wild-type petunia, which was consistent with the elongated flower phenotype (Figure 5B).

Subsequently, the F_1_ plants of the tobacco OE-31 line, including OE31-2, OE31-4, OE31-6, and OE31-8, were selected to further test whether the elongated petal phenotype could be inherited in a stable manner. As a result, a semi-quantitative PCR showed that *PhDof28* was overexpressed in all the F_1_ plants (Appendix A). Phenotypic observations indicated that the diameter and length of F_1_ transgenic tobacco flowers were significantly larger than those of the wild-type (Figure 5D), which demonstrated stable phenotype inheritance.

### 2.4. The Endogenous Hormone Contents in Transgenic Petunia

The contents of six endogenous hormones, including IAA, JA, GA, abscisic acid (ABA), zeatin (ZT), and brassinosteroid (BR) were determined in the petals of transgenic petunia plants overexpressing *PhDof28*. Results indicated that the contents of IAA in transgenic petals were increased compared to those of wild-type plants. JA and GA contents were significantly higher in transgenic petunia only at the full-flowering stage and at the semi-flowering stage separately. The transgenic petunia maintained the same ABA levels as the wild-type plants. The contents of ZT and BR decreased significantly in the transgenic plants, while both maintained a consistent level at semi- and full-flowering stages (Figure 6).

### 2.5. Expression Profile of Cell Expansion Associated Genes in Transgenic Petunia Petals

Two petal size TF inhibitors, including a bHLH TF gene, BIGPETAL (*PhBPEp*), and a NAC domain-containing protein 100 (*PhNAC100*), as well as eight cell expansion associated structural genes, including aquaporin (*PhPIP1;1*), cellulose synthase (*PhCesA5/6*), xyloglucan endotransglucosylase (*PhXTH6/22/23*), and expansion (*PhEXPA2/3*), were selected, and their expression profiles explored in transgenic petunia petals overexpressing *PhDof28* during the pre-flowering stage. As a result, the levels of gene expression of *PhBPEp* and *PhNAC100* were significantly up-regulated compared to the wild-type, while the levels of *PhEXPA2*, *PhEXPA3*, and *PhCesA6* were increased in transgenic petunia. In contrast, the levels of gene expression of *PhCesA5*, *PhXTH6*, *PhXTH22*, *PhXTH23*, and *PhPIP1;1* in the transgenic petunia were significantly down-regulated (Figure 7).

## 3. Discussion

### 3.1. Overexpression of PhDof28 Causes Petal Expansion in Petunia and Tobacco

Flower size is mainly influenced by petal cell division and expansion, and it represents one of the most vital determinants of the economic value of ornamental plants [17]. The Dof proteins are plant-specific TFs containing a C_2_-C_2_ single zinc finger structural domain that is highly conserved, and are widely participated in plant development [11]. Our recently reported Dof-type *PhDof28* TF gene [16], with preferential expression in petunia plant petals, was selected for further functional analysis to determine its roles in the petunia petal expansion process. Generally, TFs have nucleus localization and transcriptional self-activation properties [18], and those specifically expressed in plant petals are usually predicted to be associated with distinct biological processes [15,19]. Consistently, this study determined that *PhDof28* had a nuclear localization and transcriptional self-activation activity (Figure 1). Moreover, its expression was significantly up-regulated at the top part of petunia petals during flower opening (Figure 2), which confirmed its typical TF properties and potential petal expansion regulatory activity in petunias.

Stable transformation of *PhDof28* in the petunia ‘W115’ and tobacco plants produced a larger transgenic petal phenotype than in wild-type plants (Figure 3 and Figure 5), which could also be stably inherited in the F_1_ plants (Figure 4). These results demonstrated that *PhDof28* could significantly increase the cell sizes in the top parts of flower petals. Interestingly, the *Arabidopsis AtDof5.4* gene, which is an ortholog of *PhDof28* with preferential expression in the petals, inhibited petal cell proliferation and expansion, and its overexpression could significantly suppress petal expansion in transgenic *Arabidopsis* [15]. Despite having highly conserved functions during the evolutionary process, functional differentiation has been reported in some orthologous genes originating from a common ancestor [20]. For example, overexpression of *AtMYB010* could induce a variety of growth and developmental abnormalities, such as stunted growth, multi-branching, early flowering, and thin stems in *Arabidopsis*. In contrast, overexpression of its orthologous *GmMYB010* gene could not produce similar phenotypes in soybean [*Glycine max* (L.) Merr.]. In addition, overexpression of the maize (*Zea mays* L.) *MYB59* TF significantly reduced the seeds’ germination rates in transgenic tobacco plants [21]; however, overexpression of its orthologous *OsMYBAS1* gene promoted germination of transgenic rice seeds [22]. Given these reports, it is reasonable to speculate that *PhDof28* and its orthologous *AtDof5.4* gene might have undergone functional differentiation during plant evolution.

### 3.2. PhDof28 Regulates Petal Size by Triggering IAA and JA-Related Pathways

The expansion of petal cells has been reported to mainly rely on the regulation of plant hormone signaling, and the synergistic effects of endogenous plant hormones are critical determinants of plant petal size [4,5]. IAA is most extensively involved in the regulation of the development of plants by activating cell-wall-expansion-related genes, which in turn loosen and synthesize the cell wall [23,24,25]. In this study, IAA content in floral expansion stages of transgenic petunia petals overexpressing *PhDof28* were remarkably higher than that of wild-type plants (Figure 6). Moreover, transgenic petunia plants showed highly increased expression levels of two previously reported cell-wall-loosening *PhEXPA2* and *PhEXPA3* genes [26], as well as a cell wall synthesis *PhCesA6* gene [27]. Together, these results suggested that *PhDof28* overexpression could enhance the accumulation of IAA in petunia petals and promote the expression of cell-expansion-associated genes.

The mutant *Arabidopsis*, *opr3* with a defect in JA synthesis, showed pronounced petal cell expansion, which was significantly reduced after JA treatment, indicating that JA could inhibit expansion of petal cells [28]. Furthermore, the *AtBPEp* gene, which is induced by JA, could inhibit petal cell size by disrupting the petal cell expansion in *A. thaliana* [5,29]. Interestingly, JA content along with the expression of petal growth inhibitor *PhBPEp* and *PhNAC100* genes were both increased in the petals of transgenic petunias overexpressing *PhDof28* (Figure 6 and Figure 7A). On the contrary, the expression levels of cell-expansion-associated genes, which involved *PhCesA5*, *PhXTH6*, *PhXTH22*, *PhXTH23*, and *PhPIP1;1,* were all decreased (Figure 7B). These above results suggested that overexpression of *PhDof28* can accelerate petal expansion by promoting IAA-related pathways in petunias. Moreover, to prevent excessive petal growth, transgenic plants could also trigger their regulatory feedback mechanism by inducing JA accumulation, leading to increased expression of petal size inhibitor TFs, such as *PhBPEp* (Figure 8).

## 4. Materials and Methods

### 4.1. Plant Materials

The *P. hybrida* L. ‘W115’ and *Nicotiana tabacum* L. were grown in the greenhouse at the Nanjing Forestry University, Nanjing, China, at 24 °C under 15 h light/9 h dark conditions. The different parts of petunia petals (top, middle, and bottom) were collected at pre-, semi-, and full-flowering stages, then frozen and stored in liquid nitrogen at −80 °C for use in RNA extraction. Leaves of *Nicotiana benthamiana* Domin grown at 16/8 h light/dark (144 µmol m^−2^ s^−1^) and 25 ± 2 °C for 30 days were selected for subcellular localization analysis.

### 4.2. qRT-PCR Analysis

Total RNA was isolated from different parts of petal samples using the RNA Prep Pure Plant Total RNA Kit (Aidlab, Beijing, China). The Reverse Transcription Kit (TransGen, Beijing, China) was used for reverse transcription of RNA samples. The qRT-PCR was performed using TB Green™ Premix Ex Taq™ (TaKaRa, Kusatsu, Japan). Specific qRT-PCR primers were designed using Primer Premier 5.0 [30]. qRT-PCR was carried out following the method previously described [31] with three biological and technical replicates per sample. Calculation of relative gene expression levels was carried out using the 2^−ΔΔCt^ method [32].

### 4.3. Subcellular Localization and Transcriptional Self-Activation Assay

The GFP::pCAMBIA1300-PhDof28 fusion vector was constructed for subcellular localization and transferred into strain GV3101 of *Agrobacterium tumefaciens* Smith and Townsend, and then infiltrated into tobacco (*N. benthamiana* Domin) leaves for instant expression to observe the transient subcellular localization of *PhDof28*. The infiltrated plants were grown for 48 h, then the injected leaves were cut into small pieces of about 0.5 × 0.5 cm, placed on glass slides, and stained with fluorescent DAPI (4′,6-diamidino-2-phenylindole) dye. The fluorescence signal was examined with an LSM710 microscope (Zeiss, Jena, Germany).

The coding sequence of the *PhDof28* gene was converted into the pGBKT7 vector for self-activation assay, and then transferred to *Saccharomyces cerevisiae* strain AH109 (WeidiBio, Shanghai, China) to acquire a self-activating yeast vector. The transformed yeast cells were cultivated in the dark for three days on selective media of sd/-trp, sd/-trp-ade, and sd/-trp-de + x-alpha-gal in a constant incubator at 30 °C to observe self-activation capacity of the target gene.

### 4.4. Stable Transformation of Petunia and Tobacco Plants

A pCAMBIA2300::*PhDof28* overexpression vector was constructed and transformed into the *A. tumefaciens* EHA105 strain. The vector was subsequently transferred into the 20-day-old tobacco and petunia seedlings using the leaf disc method as mentioned previously [31,33], and PCR validation was conducted, with 35S-F and PhDof28-R as primers (Appendix A). After developing more than three flowers, transgenic plants with a good phenotype were selected to measure flower diameter and length. The F_1_ plants were obtained through selfing and harvesting to continue observing their phenotype.

### 4.5. Scanning Electron Microscope (SEM) Analysis of Petal Cells

The F_0_ transgenic flowers of tobacco, petunia, and wild type plants from the top, middle, and bottom parts were selected, then cut into 0.5 × 0.5 cm^2^ sections (Figure 2). This process was completely carried out in the formaldehyde-acetic-acid-ethanol fixative to avoid oxidation of petal tissues. The cellular morphologies of petals were then observed under 300×, 500×, 600×, 1000×, and 2000× magnifications using SEM (JEOL, Tokyo, Japan).

### 4.6. Measurement of Hormone Content

To quantify hormone content, wild-type and F_1_ transgenic petunia petals at the semi- and full-flowering stages were separately gathered in four biological replicates. Samples were ground in liquid nitrogen, then extracted with an extraction buffer prior to measurement of hormone content in petals with the ELISA method [34,35].

### 4.7. Screening for Petal Candidate Genes Associated with Cell Expansion

To explore the function of *PhDof28* in regulating the petal size, the cDNA sequences of previously reported structural genes and TFs potentially regulating petal size were retrieved from the TAIR (https://www.arabidopsis.org/, accessed on 10 July 2023) and the National Center for Biotechnology Information (NCBI) database (https://www.ncbi.nlm.nih.gov/, accessed on 10 July 2023). Then, the full-length amino acid sequences were used to probe their corresponding orthologous members in the petunia genome database (https://solgenomics.net/organism/Petunia_axillaris/genome, accessed on 10 July 2023). In addition, qRT-PCR primers were designed to explore the expression levels of retrieved candidate genes in the F_1_ transgenic petunia petal at the pre-flowering stage (Appendix A).

### 4.8. Data Analysis

The mean ± standard error (SE) of at least three biological replicates were calculated for all data results. The data in this study were statistically analyzed using independent samples t-test or one-way ANOVA in SPSS, and the variability of the data was inferred by Duncan’s multiple range test at the *p* < 0.05 * or *p* < 0.01 ** level.

## 5. Conclusions

This study identified the nucleus-localized transcriptional activator *PhDof28* gene, which showed preferential expression at the top parts of petunia flower petals. Transgenic tobacco and petunia ‘W115’ plants overexpressing *PhDof28* could enhance their petal sizes by promoting the expression levels of cell-expansion-related genes, such as *PhEXPA2*, *PhEXPA3*, and *PhCesA6*. Interestingly, the expression levels of some cell-expansion-related genes, including *PhCesA5*, *PhXTH6*, *PhXTH22*, *PhPIP1;1*, and *PhXTH23,* were inhibited, while the petal expansion inhibitor TFs, *PhNAC100* and *PhBPEp*, were significantly up-regulated. In addition, increased levels of plant endogenous IAA and JA hormones, which induce and inhibit petal elongation, respectively, were observed in transgenic plant petals. These results indicated that *PhDof28* overexpression could increase the rate of petal elongation by promoting the accumulation of IAA. In contrast, to avoid excessive petal elongation, transgenic plants triggered a JA-related feedback regulation mechanism by enhancing the expression of petal inhibitor genes. This study will not only help deepen the comprehension of the functions of Dof TF family members in the regulation of plant ornamental traits, but also provide new genetic resources for future molecular breeding and improvement of petunias.

## Figures and Tables

**Figure 1 ijms-24-11999-f001:**
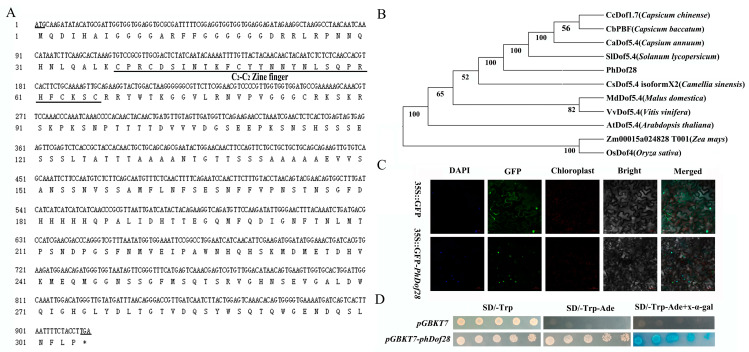
Sequence pcharacteristics of *PhDof28*. (**A**) The *PhDof28* cDNA and its encoded amino acid sequences showing the conserved Dof domain; (**B**) The phylogenetic tree of PhDof28; (**C**) GFP fluorescent signals showing the nucleus localization of PhDof28; (**D**) The transcriptional self-activation analysis of *PhDof28*. * Means the termination codon.

**Figure 2 ijms-24-11999-f002:**
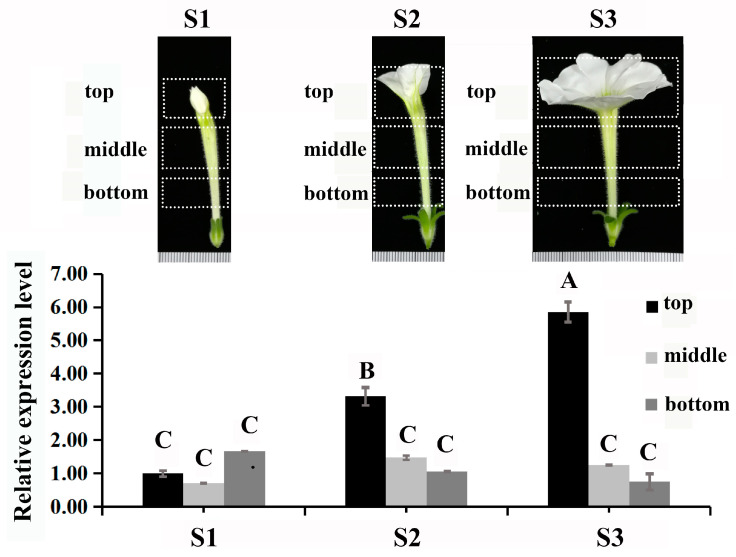
Expression levels of *PhDof28* in different parts of petunia ‘W115’ at three stages of petal development. S1: Pre-flowering stage; S2: Semi-flowering stage; S3: Full-flowering stage. The data are mean values of three biological and three technical replicates. The error bar represents SE; alphabetical letters above the bars denote significant differences between groups at *p* < 0.01 (Duncan’s test).

**Figure 3 ijms-24-11999-f003:**
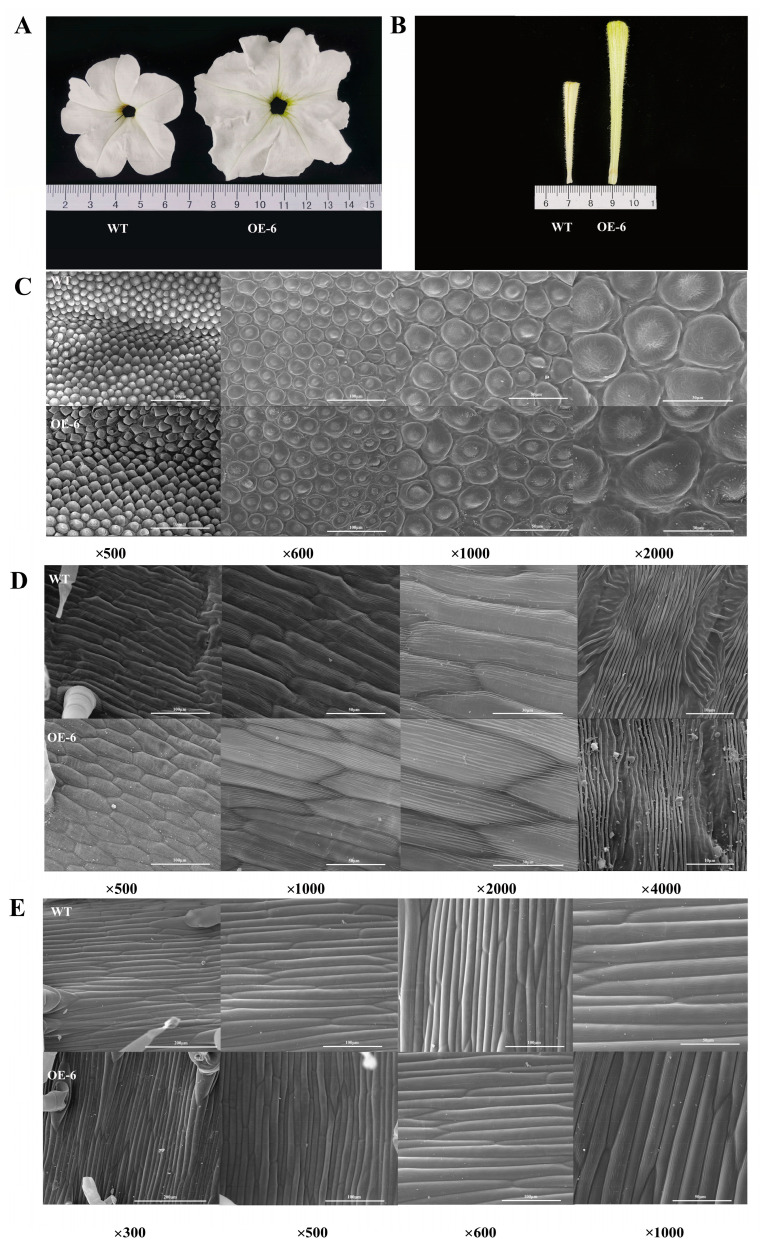
Phenotypic observation of petunia petals overexpressing *PhDof28*. (**A**) The top part of the petal; (**B**) Middle and bottom parts of the petal; (**C**) SEM (Scanning Electron Microcopy) observations of the top part of the petal; (**D**) SEM observations of middle part of the petal; (**E**) SEM observations of the bottom part of the petal.

**Figure 4 ijms-24-11999-f004:**
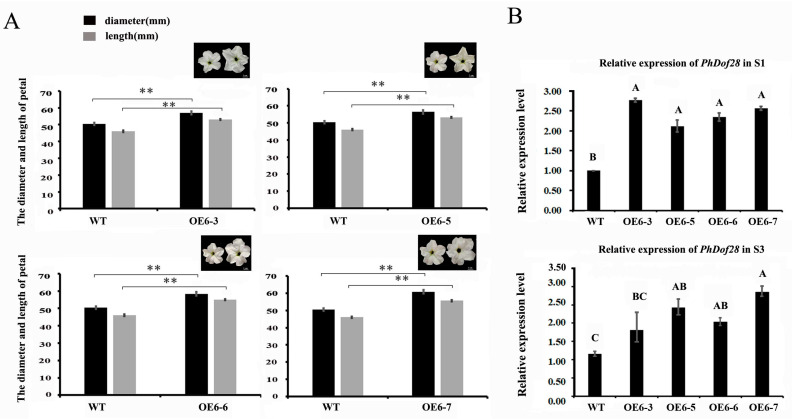
Petal phenotypes and expression levels of *PhDof28* in F_1_ transgenic petunia. (**A**) Petal diameter and flower length of the F_1_ transgenic petunia; (**B**) Expression levels of *PhDof28* in transgenic petunia petals at S1 and S3 stages. S1: pre-flowering stage, S3: semi-flowering stage. The data are mean values of three biological and three technical replicates. The error bar represents the SE. The asterisks and the alphabetical letters denote the significant difference between values at *p* < 0.01 (Duncan’s test).

**Figure 5 ijms-24-11999-f005:**
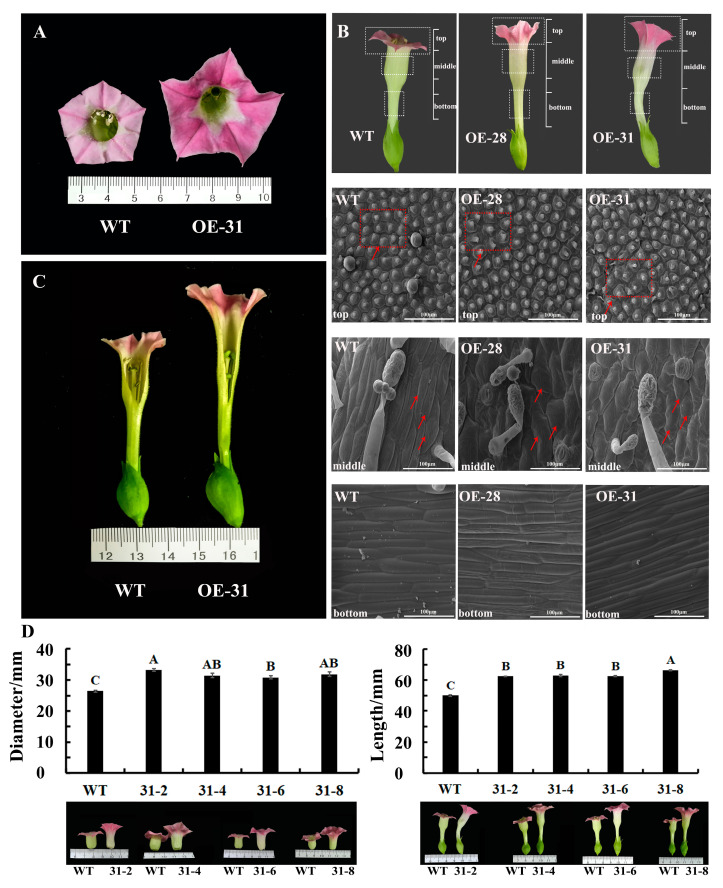
Phenotypic analysis of petals and SEM observation of petal cells from transgenic tobacco. (**A**) The front diameter of wild-type and OE-31 flowers; (**B**) SEM images of petal cells at the top, middle, and bottom parts of transgenic OE-31 and OE-28 tobacco plants. The red boxes and arrows indicate different petal cell morphology; (**C**) Sections of wild type and OE-31 flowers; (**D**) Statistical summary of F_1_ transgenic tobacco phenotypes. The alphabetical letters above the bars denote significant differences between groups at *p* < 0.01 (Duncan’s test).

**Figure 6 ijms-24-11999-f006:**
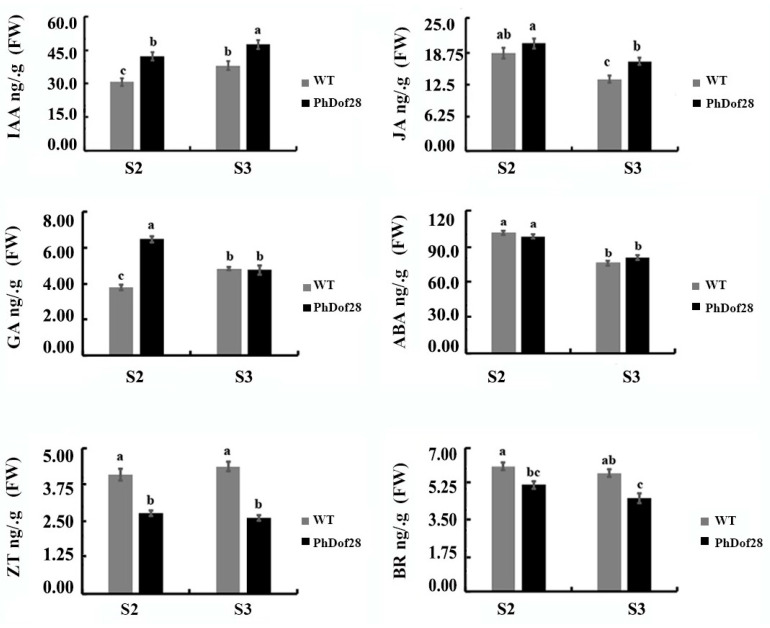
The contents of six endogenous hormones in the transgenic and wild-type petunia petals at S2 and S3 stages. S2: full-flowering stage, S3: semi-flowering stage. The data are mean values of four biological and three technical replicates. The error bar represents SE. Alphabetical letters above the bars denote the significant difference between groups at *p* < 0.05 (Duncan’s test).

**Figure 7 ijms-24-11999-f007:**
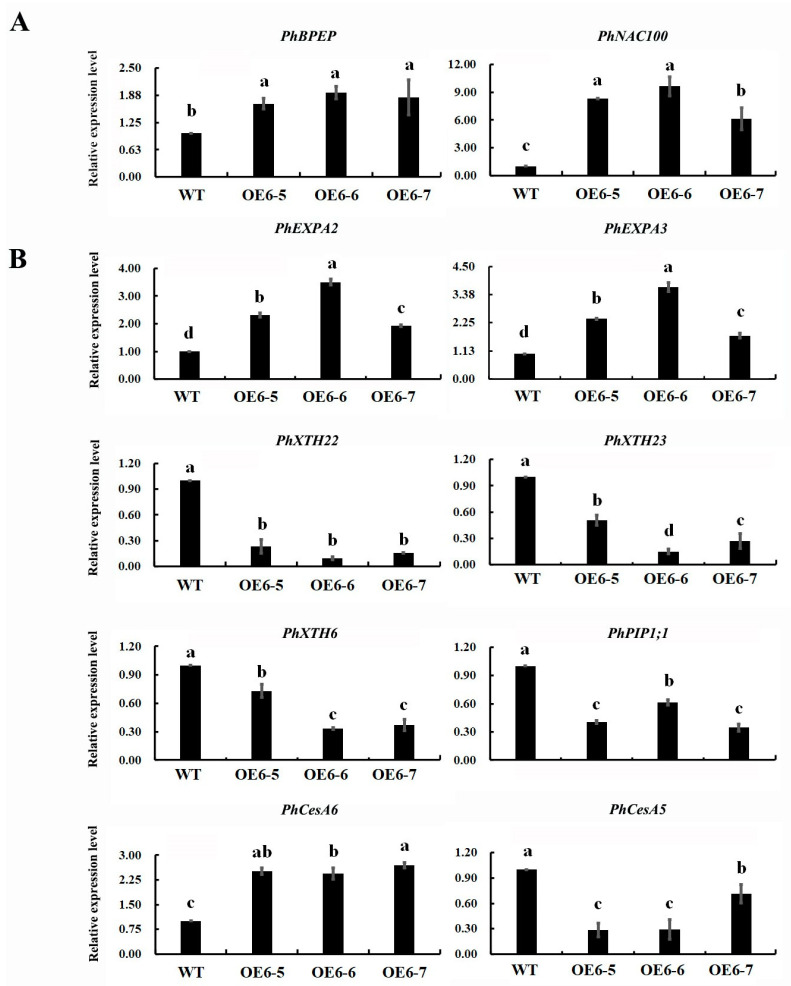
Expression levels of cell expansion related genes in the transgenic petunia petals. (**A**) The expression levels of *PhBPEp* and *PhNAC100* TFs; (**B**) the expression levels of eight cell expansion-related structural genes. The data are mean value of three biological and technical replicates. The error bar represents SE. Alphabetical letters above the bars denote significant difference between groups at *p* < 0.05 (Duncan’s test).

**Figure 8 ijms-24-11999-f008:**
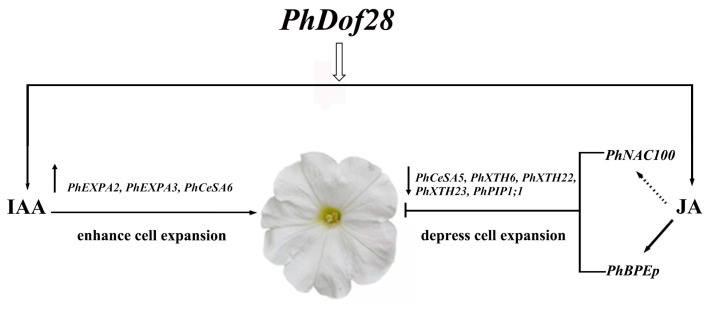
Hypothetical model illustrating the role of *PhDof28* in the regulation of petal size in petunias. The overexpression of *PhDof28* accelerated the content of IAA in petunia petals, and promotes the expression of cell-expansion-related genes, *PhEXPA2*, *PhEXPA3,* and *PhCeSA6*. Moreover, to prevent excessive petal growth, transgenic plants could also trigger their regulatory feedback mechanism by increasing JA accumulation, leading to increased expression of *PhBPEp*; at the same time, *PhNAC100*, *PhXTH6*, *PhXTH22*, *PhCeSA5*, *PhXTH23* and *PhPIP1;1* were down-regulated to depress cell expansion.

## Data Availability

All data in this study can be found in the manuscript or Appendix A.

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
