# Peer review of "DNA-Binding One Finger Transcription Factor PhDof28 Regulates Petal Size in Petunia"

_ijms, 2023, doi:10.3390/ijms241511999_

Round 1
Reviewer 1 Report
This is an interesting manuscript, but you may improve this article in order to publish in this journal. Otherwise, I have a lot of recommendations to increase the quality of your paper. Be careful with the writing and mistakes.
First of all you have to follow the rules of this journal. It is highly recommended to put the numbers of every line in the right part of your manuscript in order to point every recommendation.
There are two keywords repeated in the article title. The keywords are “Petunia” and “Petal size”. In order to increase the visibility of your paper I recommend changing these keywords. If you change them by other keywords, you will increase the probability that your paper could be found by future readers when they look for your paper in some databases like Scopus for example. If you repeat the same words in the article title and in keywords, less people could find your work. So, you must think about the visibility of your research.
When you write an acronym you must write in capitals the letters that you use to build it, for example, in the abstract you must write as follows: “Transciption Factor (TF)”. This simple thing will make the reading easier for potential and future readers.
When you write a scientific name, you must write the authors, so you must write the authors of “Petunia hybrida”.
When you write a scientific name, you must write the authors, so you must write the authors of “Arabidopsis thaliana”.
When you write a scientific name, you must write the authors, so you must write the authors of “Rosa hybrida”. This is a very common mistake in your whole paper, please, fix it.
When you write a scientific name, you must write the authors, so you must write the authors of “Chrysanthemum morifolium”. This is a very common mistake in your whole paper, please fix it.
You must use scientific names when you write a common name of a species because this is a scientific journal. So, you must write the scientific names of apple, tobacco, bamboo, maize, soy bean…
The Figure 1 if very difficult to read because the size of the letters is so tiny.
On page 4 you write SEM by the first time, so you must write its meaning into brackets as follows: “SEM (Scanning Electron Microcopy)”.
Another important thing is that you must write in capitals the letters that you have used to build the acronym, so in Figure 3 you must write as follows: “Scanning Electron Microscopy (SEM)”.
Just at the end of the page 6 you have to delete the point just before “(Figure 6)”. This point is obviously a mistake.
The letters, which are written inside the colored boxes of the Figure 8, are very difficult to read, please fix it.
When you write a scientific name, you must write the authors, so you must write the authors of “Nicotiana tabacum”. This is a very common mistake in your whole paper, please, fix it.
When you write a scientific name, you must write the authors, so you must write the authors of “Nicotiana benthamiana”. This is a very common mistake in your whole paper, please, fix it.
When you write a scientific name, you must write the authors, so you must write the authors of “Agrobacterium tumefaciens”. This is a very common mistake in your whole paper, please, fix it.
A very important question is that you must write in italics all the scientific names, so, you must write Saccharomyces cerevisiae in italics. This is scientific journal, so you must follow this rule.
You must write as well the meaning of NCBI.
Otherwise, the authors adequately developed the Introduction, presenting the problems but you must write explicitly the objectives of this paper.
The methods are adequate.
The Discussion is well developed and the data presented are correctly compared with other papers.
The authors are to be congratulated for the results obtained in this article.
Your English is very well.
Author Response
This is an interesting manuscript, but you may improve this article in order to publish in this journal. Otherwise, I have a lot of recommendations to increase the quality of your paper. Be careful with the writing and mistakes.
Response: Thanks for your great guideline. We respect your comments and have dealt with each point seriously in the following responses.
First of all you have to follow the rules of this journal. It is highly recommended to put the numbers of every line in the right part of your manuscript in order to point every recommendation.
Response: Thanks for your suggestion. The numbers of every line have been added in the right part of this manuscript.
There are two keywords repeated in the article title. The keywords are “Petunia” and “Petal size”. In order to increase the visibility of your paper I recommend changing these keywords. If you change them by other keywords, you will increase the probability that your paper could be found by future readers when they look for your paper in some databases like Scopus for example. If you repeat the same words in the article title and in keywords, less people could find your work. So, you must think about the visibility of your research.
Response: Thanks for your guideline. The keywords “Petunia” and “Petal size” have been changed to “Ornamental plant” and “Petal elongation”, respectively. (Line 30)
When you write an acronym you must write in capitals the letters that you use to build it, for example, in the abstract you must write as follows: “Transciption Factor (TF)”. This simple thing will make the reading easier for potential and future readers.
Response: Thanks for your suggestion. We have checked our manuscript. The “Transciption factor (TF)” has been changed to “Transciption Factor (TF)”. (Line 15 and 55)
When you write a scientific name, you must write the authors, so you must write the authors of “Petunia hybrida”.
When you write a scientific name, you must write the authors, so you must write the authors of “Arabidopsis thaliana”.
When you write a scientific name, you must write the authors, so you must write the authors of “Rosa hybrida”. This is a very common mistake in your whole paper, please, fix it.
When you write a scientific name, you must write the authors, so you must write the authors of “Chrysanthemum morifolium”. This is a very common mistake in your whole paper, please fix it.
Response: Thanks for your guideline. The scientific name of “Petunia hybrida” has been changed to “Petunia hybrid L.” (Line 13 and 33). The scientific name of “Arabidopsis thaliana” has been changed to “Arabidopsis thaliana L.” (Line 59). The scientific name of “Rosa hybrida” has been changed to “Rosa hybrida L.” (Line 64). The scientific name of “Chrysanthemum morifolium” has been changed to “Chrysanthemum morifolium Ramat.” (Line 68).
You must use scientific names when you write a common name of a species because this is a scientific journal. So, you must write the scientific names of apple, tobacco, bamboo, maize, soy bean…
Response: Thanks for your guideline. The scientific names of apple, tobacco, bamboo, maize, soy bean have been added. (Line 74, 76, 230, 229 and 293)
The Figure 1 if very difficult to read because the size of the letters is so tiny.
Response: Thanks for your advice. The size of the letters in Figure 1 has been enlarged.
On page 4 you write SEM by the first time, so you must write its meaning into brackets as follows: “SEM (Scanning Electron Microcopy)”.
Response: Thanks for your help. The “SEM” has been changed to “SEM (Scanning Electron Microcopy)”. (Line 120)
Another important thing is that you must write in capitals the letters that you have used to build the acronym, so in Figure 3 you must write as follows: “Scanning Electron Microscopy (SEM)”.
Response: Thanks for your help. The “SEM” has been changed to “Scanning Electron Microscopy (SEM)”. (Line 127)
Just at the end of the page 6 you have to delete the point just before “(Figure 6)”. This point is obviously a mistake.
Response: Sorry for this mistake. The point has been deleted. (Line 183)
The letters, which are written inside the colored boxes of the Figure 8, are very difficult to read, please fix it.
Response: We gratefully appreciate for your valuable suggestion. The colored boxes have been removed in Figure 8.
When you write a scientific name, you must write the authors, so you must write the authors of “Nicotiana tabacum”. This is a very common mistake in your whole paper, please, fix it.
When you write a scientific name, you must write the authors, so you must write the authors of “Nicotiana benthamiana”. This is a very common mistake in your whole paper, please, fix it.
When you write a scientific name, you must write the authors, so you must write the authors of “Agrobacterium tumefaciens”. This is a very common mistake in your whole paper, please, fix it.
Response: Thanks for your guideline. The scientific name of “Nicotiana tabacum” has been changed to “Nicotiana tabacum L.”(Line 279). The scientific name of “Nicotiana benthamiana” has been changed to “Nicotiana benthamiana Domin”(Line 283). The scientific name of “Agrobacterium tumefaciens” has been changed to “Agrobacterium tumefaciens Smith and Townsend”(Line 296).
A very important question is that you must write in italics all the scientific names, so, you must write Saccharomyces cerevisiae in italics. This is scientific journal, so you must follow this rule.
Response: Thanks for your guideline. The scientific name of “Saccharomyces cerevisiae” has been written in italics. (Line 304)
You must write as well the meaning of NCBI.
Response: Thanks for your great guideline. The “NCBI” has been changed to “National Center for Biotechnology Information (NCBI)”.(Line 343)
Otherwise, the authors adequately developed the Introduction, presenting the problems but you must write explicitly the objectives of this paper.
Response: Thanks for your great suggestion. The sentence “The purpose of this study was to further functionally characterize PhDof28 and determine its roles in the petal cell expansion.” has been changed to “The purpose of this study was to further functionally characterize PhDof28 roles in the petal cell expansion and supply gene resource for the petal size improvement breeding of ornamental plants.” (Line 84)

Reviewer 2 Report
The manuscript “A DNA-binding One Finger Transcription Factor PhDof28 Regulates Petal Size in Petunia” functionally characterizes the newly identified transcription factor, PhDof28 in petunia, and this study reveals the antagonistic effect of plant hormone IAA and JA during the flower expansion process. PhDof28 overexpression lines exhibit larger petal phenotypes in both petunia and tobacco plants.
Minor suggestions
-
Page 1, Keywords: do not need to capitalize all the words
-
Page 10, Figure 8 legend. Line 3, PhEXPA2, PhEXPA3 and PhCeSA6, please put commas in between genes.
-
Page 10, Figure 8 legend. Line 6, PhNAC100, PhXTH6, PhXTH22, PhCeSA5, PhXTH23, and PhPIP1;1 please put commas instead of “、” in between these genes.
-
Page 10, 4.3 Subcellular localization and transcriptional self-activation assay section, please italicize Saccharomyces cerevisiae
Major concerns
-
For the supplementary figure 1, PCR positive detection of F0 generation transgenic petunia. What are the reasons that the PCR product bands are not consistent? Some transgenic lines have stronger bands (OE#4-7) compared to (OE#1-3).
- For supplementary figure 2, what does the label D mean? Please specify?
Author Response
The manuscript “A DNA-binding One Finger Transcription Factor PhDof28 Regulates Petal Size in Petunia” functionally characterizes the newly identified transcription factor, PhDof28 in petunia, and this study reveals the antagonistic effect of plant hormone IAA and JA during the flower expansion process. PhDof28 overexpression lines exhibit larger petal phenotypes in both petunia and tobacco plants.
Minor suggestions
Page 1, Keywords: do not need to capitalize all the words.
Response: Thanks for your great guideline. We have corrected the format of the keywords as required. (Line 30)
Page 10, Figure 8 legend. Line 3, PhEXPA2, PhEXPA3 and PhCeSA6, please put commas in between genes.
Response: Thanks for your great guideline. We have put commas in between genes as suggested. (Line 277)
Page 10, Figure 8 legend. Line 6, PhNAC100, PhXTH6, PhXTH22, PhCeSA5, PhXTH23, and PhPIP1;1 please put commas instead of “、” in between these genes.
Response: Thanks for your help. We have put commas instead of “、” in between PhNAC100, PhXTH6, PhXTH22, PhCeSA5, PhXTH23, and PhPIP1;1 as required. (Line 280)
Page 10, 4.3 Subcellular localization and transcriptional self-activation assay section, please italicize Saccharomyces cerevisiae
Response: Thanks for your great guideline. The scientific name of “Saccharomyces cerevisiae” has been written in italics. (Line 304)
Major concerns
For the supplementary figure 1, PCR positive detection of F0 generation transgenic petunia. What are the reasons that the PCR product bands are not consistent? Some transgenic lines have stronger bands (OE#4-7) compared to (OE#1-3).
Response: Thanks for this great question. It is indeed that the transgenic lines OE#4-7 have stronger bands compared to OE#1-3. We think the result is mainly due to the different concentrations of DNA templates. The DNA extractions of transgenic lines OE#4-7 and OE#1-3 were conducted by different students because of the pandemic. As the result has already showed the positive situation of transgenic petunia, so we didn’t adjust the concentrations of DNA samples and perform the PCR detection again. Thanks for this reminding, we will pay more attention to this problem in our following study.
For supplementary figure 2, what does the label D mean? Please specify?
Response: Thanks for your great suggestion. Label D means the ddH2O was used as the template of blank control. The sentence “Label D means the ddH2O was used as the template of blank control.” has been added in the legend supplementary figure 2.
Thanks for your careful reading and fair evaluation of our work. We respect your comments and have dealt with each comment seriously.
